# From work stress to disease: A computational model

**Remco Benthem de Grave**⍟¤*, **Fred Hasselman, Erik Bijleveld**⍟

Behavioural Science Institute, Radboud University, Nijmegen, The Netherlands

¤ Current address: Open Lab, School of Computing, Newcastle University, Newcastle upon Tyne, United Kingdom
* r.benthemdegrave2@newcastle.ac.uk

**Data Availability Statement:** All relevant data are within the paper and its Supporting information files.

**Funding:** EB was supported by grant 016-165-100 from the Nederlandse Organisatie voor Wetenschappelijk Onderzoek (https://www.nwo.nl/en). The funders had no role in study design, data

## Abstract

In modern society, work stress is highly prevalent. Problematically, work stress can cause disease. To help understand the causal relationship between work stress and disease, we present a computational model of this relationship. That is, drawing from allostatic load theory, we captured the link between work stress and disease in a set of mathematical formulas. With simulation studies, we then examined our model's ability to reproduce key findings from previous empirical research. Specifically, results from Study 1 suggested that our model could accurately reproduce established findings on daily fluctuations in cortisol levels (both on the group level and the individual level). Results from Study 2 suggested that our model could accurately reproduce established findings on the relationship between work stress and cardiovascular disease. Finally, results from Study 3 yielded new predictions about the relationship between workweek configurations (i.e., how working hours are distributed over days) and the subsequent development of disease. Together, our studies suggest a new, computational approach to studying the causal link between work stress and disease. We suggest that this approach is fruitful, as it aids the development of falsifiable theory, and as it opens up new ways of generating predictions about why and when work stress is (un)healthy.

## Introduction

Work stress has been estimated to cost between $221 to $187 billion annually [1]. Considering these high societal costs, it is not surprising that scientists have thoroughly studied the origins, the nature, and the consequences of work stress. Broadly, work psychologists have examined how and when work stressors shape well-being and performance; biological psychologists have examined the nature of the physiological stress response; and, epidemiologists have examined how and when work stress may cause disease on the long run. Yet, despite the maturity of the science of work stress, this important area has a clear shortcoming: it does not yet have computational models to represent how work stress may cause disease.

*Computational models* can advance scientific knowledge in various ways. For example, they contribute to the transparency and falsifiability of theory, they facilitate the understanding of

collection and analysis, decision to publish, or
preparation of the manuscript.

**Competing interests:** The authors have declared
that no competing interests exist.

potential causal mechanisms, and they help generate new predictions [2–4]. In what follows, drawing from *allostatic load theory* [5], we propose a computational model of the putative causal effect of work stress on disease. In turn, we test whether our model can explain core findings from several previous empirical studies. Finally, we use our model to generate new predictions about the relationship between workweek configurations (i.e., how work hours are distributed over the working week) and the development of disease. Together, our research uses computational modelling to address a central problem in the domain of occupational health: how does work stress affect people's health?

## A brief introduction to computational models

This section is intended for readers who are new to computational models. Readers who are familiar with computational modeling may skip this section.

We will start explaining what computational models are, by comparing them to *verbal theories* (with *theory* synonymous for *model*). By *verbal theories*, we refer to theories not described through mathematical equations (nor through some other structure of formal logic). Most theories in the social sciences are verbal theories. An example of a well-known verbal theory is *cognitive dissonance theory* [6]. Cognitive dissonance theory predicts that attitude change takes place when people's previously-held attitudes are inconsistent with their behavior. Another well-known verbal theory in social sciences is *social facilitation theory* [7, 8], which predicts that, when in presence of others, people perform better on well-learned and simple tasks, but worse on new and complex tasks [9]. What these theories have in common is that they are defined in words, not in mathematical equations. As a result, they lack specificity. Little is clear about, say, the dynamics of how attitude change develops over time. Is the speed of change constant? Or does it increase first and then plateau? Or, how about the relationship between the presence of other people and performance? Is this relationship linear? What conditions are required for this relationship to hold? In order to perform any quantitative test of a verbal theory, e.g., using statistical analyses, researchers always need to make several interpretations of the theory first, leaving much room for flexibility. It is thus difficult, and perhaps impossible, to rigorously test verbal theory [10].

In physics and engineering, most (if not all) theories are computational. An example of a computational theory is Newton's theory of universal gravitation. This theory explains the force that works between two masses with the equation $F = G(m_1 m_2)/r^2$, in which $m_1$ and $m_2$ are the two masses, $r$ the distance between the two masses, $G$ the universal gravitation constant, and $F$ the resulting force. Each combination of values that are inserted for the parameters in the right side of the equation result in a single, exact, resulting force. There is transparency in the dynamics that describe the relationship between parameters, limiting the assumptions that researchers have to make. Thus, computational models can typically be tested with greater rigor.

**Benefits of computational modeling.** First, more so than verbal models, computational models are falsifiable. As mentioned, the limited specificity of verbal theory leaves room for various ways in which to define relationships between parameters. This flexibility makes it difficult to falsify verbal theories [11, 12], thus limiting what can be learned from a study that attempts to empirically validate a verbal theory. In contrast, computational models are fully specific. The model provides transparency about the included parameters and the assumed relationships between them, thus limiting interpretational freedom, facilitating falsifiability [3, 13, 14].

Second, computational models can help the understanding of underlying causal mechanisms of the relationship that is being studied. Rather than attempting to create a model that

exactly mimics reality, the goal of modeling is to provide a model that can satisfyingly approximate empirical observations, while maintaining simplicity as much as possible [2, 15]. As such, modeling can give insight in the dynamics that govern the process under investigation. Computational models may also increase understanding through analogies. In particular, processes that seem unrelated can have models that are computationally the same (e.g., the same model may explain the behavior of both arteries in the body, and the behavior of rubber bands that hold objects together). This way, researchers can borrow principles from well-developed theories from other scientific areas, accelerating scientific progress [2].

Third, computational models can be used to perform simulations, which has several important advantages. Through simulations, researchers can explore mechanisms that can explain empirical findings, e.g., [16, 17]; clarify inconsistencies among previous findings, e.g., [18–20]; examine a model's robustness, i.e., examine the parameter ranges under which a model explains existing empirical data [2]; and scrutinize the logic of intuitive reasoning behind a theory, which is sometimes flawed [21]. In other words, simulations can examine whether a theory is viable to begin with. Furthermore, simulations of computational models can also generate novel predictions of behavior that may be observed in real populations [2]. In some cases, intuitive interpretation of a theory does not lead to clear predictions. To provide an example from the present research: Does taking free days on Wednesdays and Sundays–instead of Saturdays and Sundays–influence the risk of developing disease in the long run? While it is not easy to use verbal models to formulate hypotheses about particular situations such as these, simulations of computational models can be used for this purpose.

## The present research

With this research, we aim to meet three goals. Our first goal is to develop a first computational model of the *work stress-disease* relationship. Rather than providing a detailed model, including many possible parameters involved in the process of becoming diseased, we aim to create a simple and compact model that focuses on the most important candidate mechanisms. We chose to prioritize parsimony as we believe that it will benefit interpretability of the model, and as it will limit the number of arbitrary assumptions that we will need to make. Rather than formulating a new theory from scratch, we will be using knowledge from leading, existing verbal theory (i.e., allostatic load theory) to develop our model.

Our second goal is to investigate our model's ability to reproduce previously-reported data, by using Monte Carlo simulations (i.e. simulations in which the value of a variable is determined by its previous value and random sampling from a distribution). For these simulations, we will simulate individuals for whom we will verify that their collective behavior is comparable to the behavior that is described in large-scale empirical studies. This investigation will provide us with feedback about the credibility of the model that has been developed.

Our third goal is to formulate new predictions, again by using MCMC simulations. Specifically, we aim to formulate predictions about the impact of how working hours are distributed over the working week. When future research tests our novel predictions, this would provide a transparent means to verify or falsify our model–and, subsequently, to improve–our model.

In the next sections, we will lay out our research. In Study 1, we will perform simulations based on our model, to verify that our model can produce cortisol time courses similar to those reported in the literature. In Study 2, we will describe the relationship between cortisol levels and disease, completing our computational model. In both Studies 1 and 2, through simulations, we will examine if our simulated people exhibit the same relationship between stressors and disease, as was reported in previous large-scale studies. In Study 3, we will use the model to make novel predictions.

## Study 1: From stress to cortisol

### Allostatic load theory

We base our computational model, summarized in Fig 1, on allostatic load theory [5]. This influential theory describes the physiological processes that mediate the causal relationship between stress and disease. Allostatic load theory starts out from the assumption that stressful situations disrupt the stable resting state (*homeostasis*) of physiological systems, causing an alternative equilibrium (*allostasis*) in which physiological systems adaptively respond to deal with the stressful situation. Generally, allostasis involves increased activity of the sympathetic nervous system and the hypothalamus-pituitary-adrenal axis (HPA axis). These activations generally cause an increase in heart rate, a release of nutrients into the bloodstream, a suppression of digestion, as well as a number of other changes [22] (for an accessible introduction, see [23]).

Although the ability to transition into allostasis is usually considered to be a healthy adaptation–i.e., it helps people to effectively deal with stressors–allostasis does put a burden on physiological systems. In many cases, this impact is fully reversible; it causes no permanent damage. For example, an artery under high pressure may readily return to its normal, resting state. However, when allostasis occurs too frequently (Type I), when allostasis is maintained for too long (Type II), or when physiological adaptation to allostasis is inadequate (Type III), lasting damage may occur. In this paper, we refer to allostasis' reversible burden on physiological systems as *allostatic strain*, and allostasis' lasting damage as *allostatic load*. As allostatic load accumulates, *disease* may emerge.

To develop a parsimonious computational model based on allostatic load theory, we have applied a simplification: we will consider only allostatic load due to allostasis occurring too frequently (i.e., Type I). To be able to test our computational model against previous empirical data, we needed to make an additional assumption: we assume that the concentration of the hormone cortisol–which is, indeed, frequently used as a direct measure of the human stress response [24, 25]–provides an index of the current physiological response that is recruited to achieve allostasis.

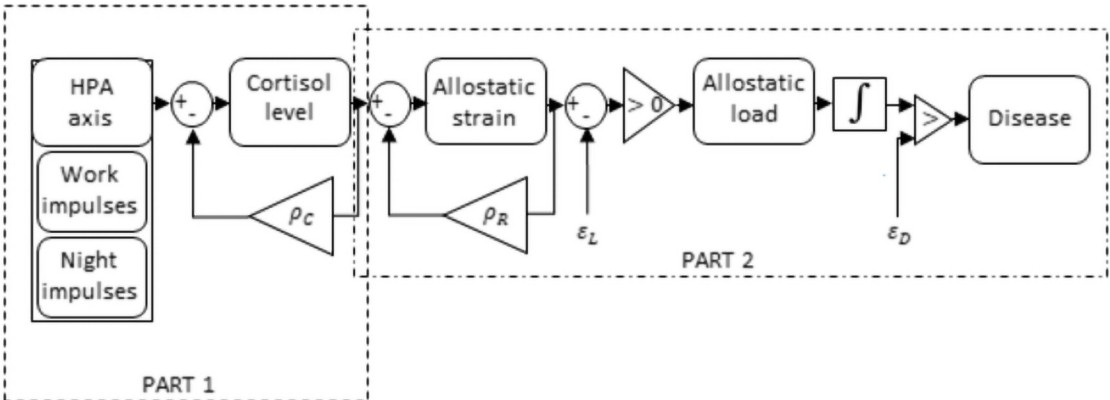

**Fig 1. Schematic illustration of our computational model.** In essence, the model explains how activation of the HPA axis, due to work stress and circadian inputs, elevates circulating cortisol levels. In turn, the cortisol response burdens the physiological system, in a fully reversible way (*allostatic strain*). Decay of cortisol and allostatic strain is assumed to be proportional to their respective current levels, as indicated by the feedback loops. When allostatic strain exceeds a threshold $\varepsilon_L$, it causes permanent damage (*allostatic load*). Allostatic load is cumulative and non-reversible (hence the integration symbol). When allostatic load exceeds a threshold $\varepsilon_D$ it ultimately causes disease. This chain of events is described in detail in the main text. HPA axis = Hypothalamus Pituitary Adrenal-axis.

**Daily fluctuations in cortisol levels.** The *HPA axis* regulates a chain of bodily events that leads to the production of cortisol [22]. The HPA axis consists of three hormonal glands: the *hypothalamus* and the *pituitary gland* (both located deep in the brain) and the *adrenal glands* (located on the kidneys). These glands produce *corticotropin-releasing hormone* (CTH), *adrenocorticotropic hormone* (ACTH) and *cortisol*, respectively. These glands operate in a cascading manner. That is, neural impulses into the hypothalamus (e.g., due to stressors) cause the hypothalamus to release CTH, which causes the pituitary gland to release ACTH, which in turn causes the adrenal glands to release cortisol. However, all three glands have receptors for their own and each other's hormonal output. Thus, they together form an intricate dynamic system, which also involves the liver, which removes cortisol from the bloodstream, and then decomposes it [26].

Bloodstream cortisol concentrations follow a characteristic circadian pattern. Generally, blood cortisol steeply rises during the last few hours of the night, peaks about half an hour after awakening (we will refer to this peak as the *morning peak*), and then gradually decreases during the day [27]. Importantly, though, individual cortisol levels are very heterogeneous [28]. One key source of variation is the strength of the response to neural impulses into the HPA axis, which differs from person to person [24]. Also, people substantially differ in the rate with which the liver decomposes cortisol: cortisol half-life varies between 60 and 90 minutes [28, 29]. Bloodstream cortisol concentrations also respond to acute stressors. After people encounter a stressor, the cortisol level peaks after about 30 minutes, after which cortisol levels decrease again [24, 30].

It is likely that the amplitude of the morning peak and strength of cortisol responses to acute stressors (during the day), are related. In particular, the morning peak may reflect the anticipation of physiological and psychological demands for that day [31, 32]. In support of this idea, research shows that people show higher morning peaks on weekdays, as compared to weekend days [33, 34]. Similarly, a study among competitive dancers showed that dancers had a larger morning peak on competition days, as compared to training days [35]. So, it is plausible that the morning peak reflects the anticipation of upcoming events, at least in part, causing a correlation between (a) the amplitude of the morning peak and (b) the strength of cortisol responses to acute stressors. In our simulations, we will consider this possibility. That is, we will run separate simulations assuming vs. not assuming this correlation.

**In a computational model.** In this part of the model we describe the relationship between neural impulses to the HPA axis, *I*, and the change in cortisol concentration, *dC/dt*. We assume that all activity of the HPA axis is the result of neural impulses to the HPA axis, which may either be the result of a circadian process taking place during the night (we will further refer to these impulses as *night impulses*), or the result of work stressors (for consistency, further referred to as *work impulses*). We assume that each impulse has a binary intensity: impulses are either happening or not happening. Due to this assumption, impulses to the HPA axis, at any specific time point, can be described as countable number, $\Sigma I_t$. Thus, a higher number of impulses at the same time can be interpreted as a stronger impulse.

We simplify the dynamics of the HPA axis by assuming that it can be described by the following equation:

$$\frac{\mathrm{d}C}{\mathrm{d}t} = -p_C C_t + \kappa_{HPA} \sum (I_t - \tau) \tag{1}$$

In this equation, a rise in the cortisol concentration that is proportional, by $\kappa_{HPA}$, to the neural impulses, $\Sigma I_t$, delayed by $\tau$. The decay of cortisol is proportional to the current cortisol concentration, $C_t$, by a decay constant, $\rho_C$.

**Simulations.** To examine the ability of the model to reproduce the cortisol patterns that were previously reported (specifically in [26, 36, 37], we performed MCMC simulations with the model, with parameters as defined in Table 1 (code available in S1 Script). As input to the model, a population of 10,000 people was simulated. Each simulated person was characterized by an individual cortisol half-life value. Specifically, for each person, a random draw from a normal distribution of cortisol half-life values determined this person's cortisol half-life, with a mean and standard deviation chosen such to reflect the variation of half-life values reported in previous literature [28, 29].

To mirror the time course of a cortisol response to an acute stressor as described in literature [24, 30], impulses always lasted 30 minutes. Each simulated person was given a unique average number of daily *night impulses* and *work impulses*.

For night impulses, this unique *average number* of daily impulses was randomly drawn from a Gamma distribution (i.e. a continuous probability distribution for values that can only be positive, such as frequency values [38], see S1 Appendix). In turn, this average number was used to randomly assign a number of impulses to each day. In particular, the number of night impulses that a person received on a particular day, was determined from a Poisson distribution (i.e., the discrete frequency distribution) with as expected frequency λ, the average number of daily night impulses for that person. For *work impulses*, we followed the same procedure.

In our simulation, we assumed work that impulses could occur at any moment within the working hours. Thus, work impulses were determined through random draws from a uniform distribution, ranging from work start to work end. Weekends were free of work impulses. We further assumed that night impulses should become exponentially more likely towards the end of the night, with the maximum probability at awakening. Thus, night impulses were randomly drawn from a negative exponential distribution (see S1 Appendix for a plot of this exponential distribution).

**Table 1. Parameter settings for simulating cortisol time courses in Study 1.**

| Parameter | Value |
|---|---|
| Sampling frequency | 2 |
| People simulated | 10,000 |
| Days simulated per person | 200 |
| Night impulses, quantity | $X \sim \Gamma(k = 3, \theta = 14)$[a] |
| Night impulses, moment | $X \sim T_{wake} - EXP(\lambda = 1)$[a] |
| Work impulses, quantity | $X \sim \Gamma(k = 3, \theta = 6)$[a] |
| Work impulses, moment | $X \sim \mathcal{U}(a = T_{ws}, b = T_{we})$ |
| Cortisol decay constant ($\rho_C$) | $X \sim \mathcal{N}(\mu = .52, \sigma^2 = .05^2)$ |
| HPA axis scaling constant ($\kappa_{HPA}$) | 2.20 |
| Cortisol response delay, $\tau$ | 30 min |
| Wake time, $T_{wake}$ | 7:00AM |
| Work start, $T_{ws}$ | 8:30AM |
| Work end, $T_{we}$ | 4:30PM |
| Working days | Mon-Fri |

[a]Parameter values were determined through visual calibration on cortisol time courses published in [37](see Fig 5a). Specifically, we adjusted these parameter values until simulations yielded a cortisol time course similar to the one reported in [37].

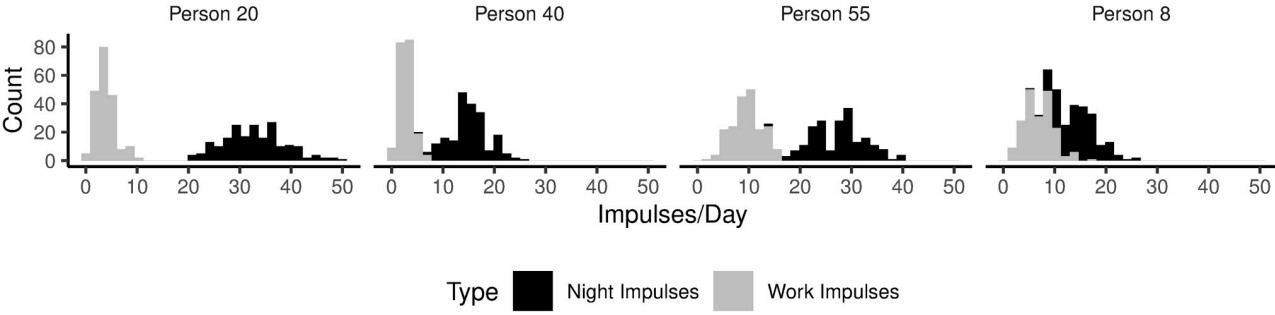

**Fig 2. The count of days (y-axis) that a simulated person received a specific number of impulses (x-axis).** Four random, representative, simulated people are shown. Note: the total amount of simulated days is 200.

To illustrate the simulation procedures laid out above, Figs 2 and 3 describe the outcomes of the procedures for four representative, simulated people. Fig 2 shows frequency distributions of night impulses and work impulses. Fig 3 shows how impulses were distributed over time during a single day. As introduced above, we ran separate simulations in which we assumed no correlation between night impulses and work impulses (Scenario I), and in which we assumed a correlation between night impulses and work impulses (Scenario II) [31]. In Scenario I, we drew average night impulses and average work impulses independently from a Gamma distribution. In Scenario II, we first drew the average number of work impulses from a Gamma distribution. Then, to define the average number of night impulses, we summed the average number of work impulses with a random value from a gamma distribution with shape parameter $k = 1$ and scale parameter $\theta = 8$. This procedure ensured (a) that the distribution for the average number of night impulses was visually similar across Scenarios I and II, and that (b) there was a correlation between average night impulses and average work impulses only in Scenario II (illustrated in Fig 4).

## Simulation results

Figs 5–7 show comparisons of cortisol time courses from the simulations with cortisol time courses from findings from the published literature. For each person, only the first day of the simulation was used to create these plots. Figs 5 and 6 only show results for Scenario I, as results for Scenario II were extremely similar (see S1 Appendix).

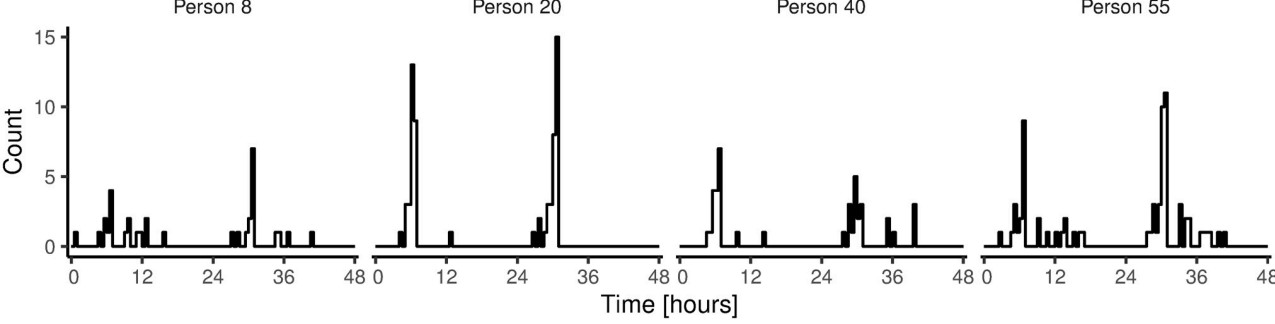

**Fig 3. Distribution of all impulses over two consecutive days (48 hours).** Two representative days are shown for each of four random, representative, simulated people. The plots illustrate that the number of impulses varies from day to day. This is because the number of impulses in a given day results from a random draw from a Poisson distribution, based on the person's overall mean (see main text).

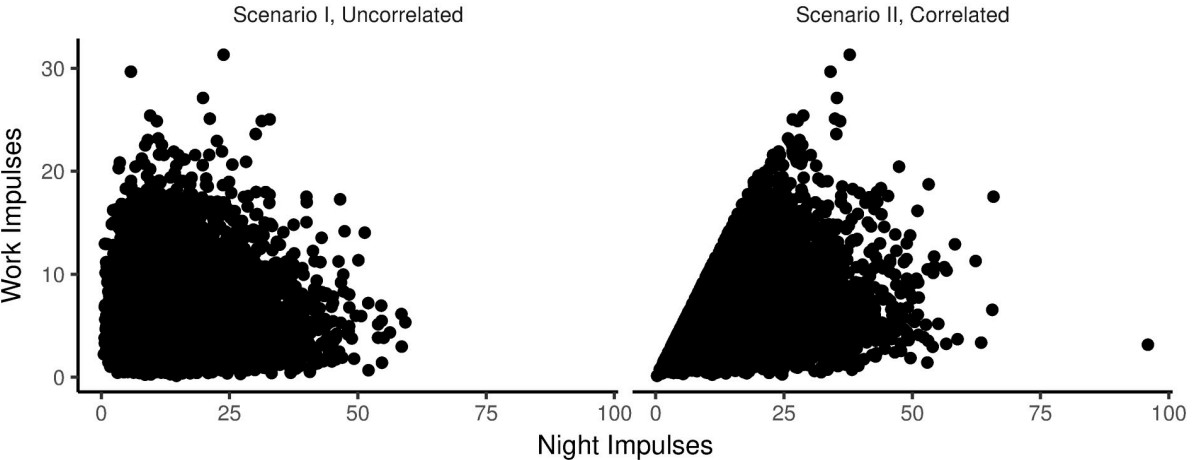

**Fig 4. Correlations between average daily work impulses and average daily night impulses.** Scenario I, uncorrelated and Scenario II, correlated.

We compared the results from our simulations to results from previous empirical studies in three ways. First, we examined our model's ability to reproduce group-level cortisol time courses during waking time. To this end, we used data from a meta-dataset from [37] as a starting point. This meta-dataset combined 15 previous field studies (total n ≈ 19,000), in order to obtain reference ranges for salivary cortisol levels in humans. Data from this previous study are replotted in Fig 5a; results from our simulations are plotted in Fig 5b. Moreover, we present more detailed comparisons between this previous meta-dataset and our simulations in S1 Fig. Specifically, for both the empirical data and our simulations, we quantified daily cortisol patterns as is often done in the literature (i.e., area under the curve with respect to the ground, $AUC_G$ [40]; area under the curve with respect to the increase, $AUC_I$ [40]; and the wake-to-bed slope [41]). We then plotted the indices based on the empirical data against the

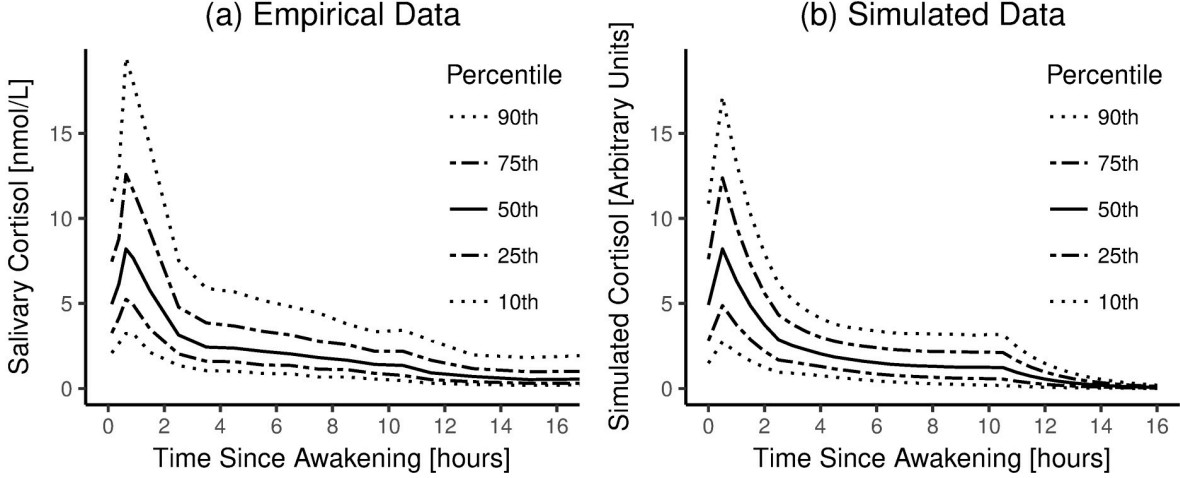

**Fig 5. Empirical vs. simulated aggregated day-time cortisol time courses.** a) Empirical data. Cortisol time courses based on data from 19,000 people, replotted from [37] with permission. b) Simulation results. Note: in both panels, data are shown until 16 hours after awakening.

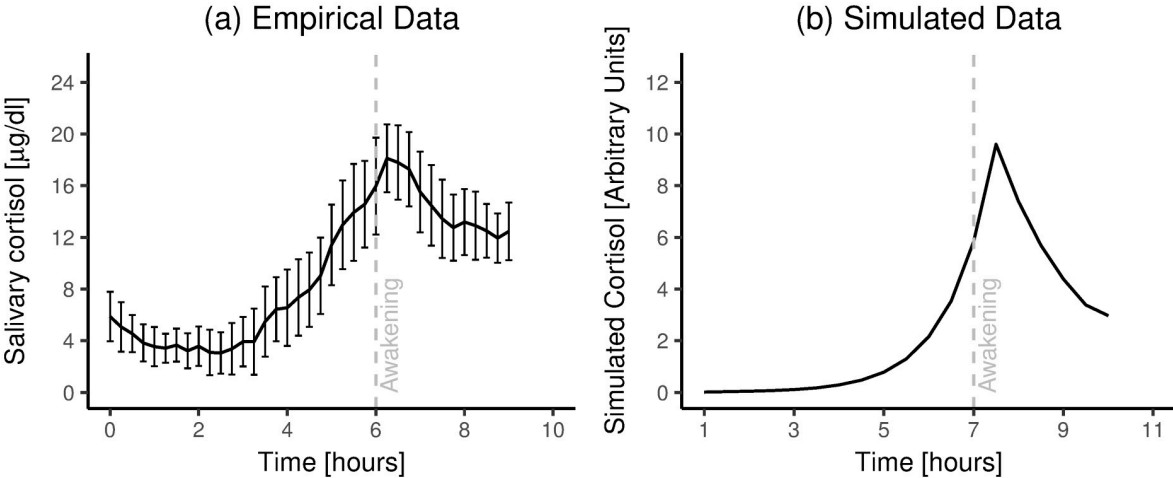

**Fig 6. Empirical vs. simulated aggregated night-time cortisol time courses.** a) Empirical data. Night-time cortisol time courses (n = 15; error bars reflect 95% confidence interval around the mean), replotted from [39] with permission. b) Simulation results. Note: in (b), the 95% confidence interval around the average is too small to be discernable.

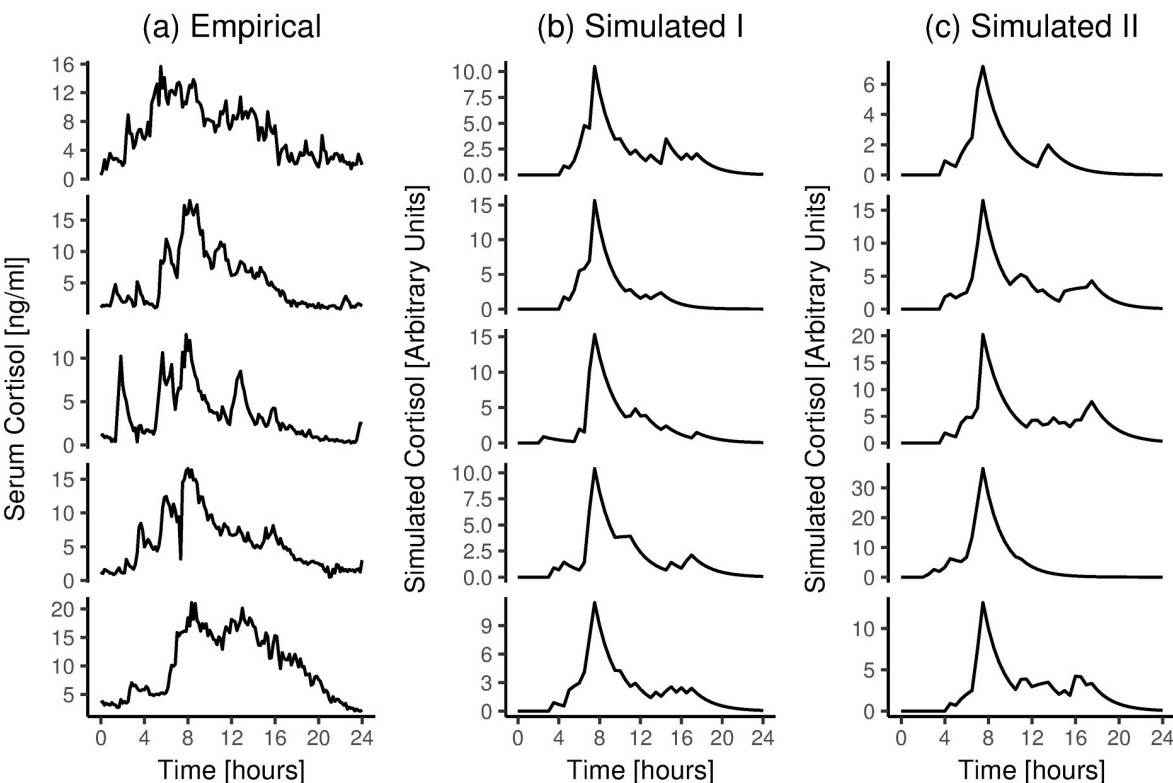

**Fig 7. Empirical vs. simulated 24-hour individual cortisol time courses.** a) Empirical data. 24-Hour cortisol time courses of five random, representative healthy individuals from [26], replotted with permission. b) Simulation results, showing five random, representative individuals based on Scenario I. c) Simulation results, showing five random, representative individuals based on Scenario II.

indices based on the simulations (S1 Fig), enabling us to further assess how well the simulations match the data, when considering well-established indices of diurnal cortisol dynamics.

Second, we examined our model's ability to reproduce group-level cortisol time courses in the hours around waking. To this end, we used data from a dataset from [36](data presented in [39]), who took blood samples from 15 participants every 15 minutes during the night and the morning, which they then assayed for cortisol. Data from this previous study are replotted in Fig 6a; results from our simulations are plotted in Fig 6b.

Third, we examined our model's ability to simulate individual-level cortisol time courses that are similar to empirical observations. To this end, we used data from a dataset from [28] (shared for public use by [26]). [28] took blood samples from people with depression (n = 12) and healthy controls (n = 17) every 10 minutes for 24 hours, starting at midnight. Data from five random, representative healthy control participants from this previous study are plotted in Fig 7a; representative results from our simulations are plotted in Fig 7b and 7c.

## Discussion

In sum, the model simulations suggest that our model is able to reproduce three key sets of empirical findings [26, 37, 39]. In particular, by applying our model, we could successfully reproduce empirically-observed cortisol concentrations, both during the night and during the day, both on the group and the individual level. Thus, at this stage, our model passes the bar to move on to the next step–that is, we conclude that we can build on this model to examine if we can predict the occurrence of disease from work stress.

Nevertheless, we found three slight differences between the empirical data and the simulation results. First, whereas empirical data suggest that cortisol levels never truly approach zero, this does happen in the simulation results (see Figs 5a vs. 5b; 6a vs. 6b; 7a vs. 7b and 7c). Perhaps, this discrepancy stems from the way we simulated work impulses. In particular, for *all* simulated people, work always ended 9.5 hours after awakening; we modeled no work impulses after that. By contrast, in life, many people remain active well beyond 9.5 hours after awakening, which may expose them to stressors (which may or may not be related to work). Such evening- and night-time activity can explain why measured cortisol levels, at least on the group level, stay above zero throughout the night. Alternatively, it may be the case that some homeostatic mechanism blocks the degradation of cortisol, after cortisol concentrations fall below some threshold. We are not aware of the existence of such a homeostatic mechanism– but if it exists, it may explain the slight discrepancy between the data and the model simulations.

Second, when we examine the group-level cortisol time courses in the hours around waking [39], it seems that our simulation results (Fig 6b) suggest a somewhat steeper rise and fall than the empirical data (Fig 6a). However, the simulated time course does seem to fall within the 95% confidence interval of the empirical data (Fig 6a). On a macroscopic level, the two plots are similar.

Third, when comparing individual-level cortisol time courses from empirical data and our simulations (Fig 7a vs. 7b), we note that there is more high-frequency variation in cortisol levels in real empirical data (i.e., lines in Fig 7a look more jagged). This difference can be explained from the fact that we used a relatively low sampling rate in our simulations. Specifically, in our simulations, stressors always had a duration of 30 minutes, whereas the duration of real-life stressors may vary considerably; they can be shorter than 30 minutes. Moreover, the measurement process that yielded the real-life data (i.e., participants had a catheter in their forearm; blood samples were taken every 10 minutes [28]) may have added some noise by itself. Although this difference between the real data vs. the simulated data may slightly distort

variance around the effect sizes reported in later sections of this paper, we expect that the estimates themselves should not be biased by our simulations having less high-frequency noise.

## Study 2: From cortisol to disease

Study 2 explains how cortisol dynamics can lead to disease. Like in Study 1, we compare our model simulations to previously-reported empirical data. Finally, we test the robustness of the predictions for variation in parameter values.

### Background

High cortisol concentrations can cause disease via several routes. That is, each organ system is affected in its own way, and even within each organ system, there are often multiple ways in which cortisol can cause damage. A detailed review of all of these routes is beyond the scope of this article; however, in what follows, we will discuss three well-established core pathways. Then, we capture the essence of these pathways in our computational model.

First, cortisol can cause disease through its actions on the cardiovascular system. Cortisol triggers a rise in blood pressure, heart rate, and cardiac output [22, 42, 43]. These cardiovascular changes help people deal with stressors and they are fully reversible (*allostatic strain*). Yet, when they are prolonged, these cardiovascular changes increase the risk of vascular lesions, which in turn promote the buildup of arterial plaque at the lesion sites (*allostatic load* [44]). These plaques harden and clog the arteries, which leads to a disease state called *atherosclerosis* [44–46]. Atherosclerosis causes symptoms depending on the location of the artery that is affected. For example, potential symptoms include shortness of breath, trouble speaking, dizziness, pain, and nausea [47]. Moreover, plaques can get loose, after which they can get stuck in narrower arteries, where they block blood supply to the distal tissue [48]. This process is potentially lethal, especially when it takes place in the heart (*myocardial infarction*) or the brain (*stroke*).

Second, cortisol can cause disease through its actions on the immune system. As a part of the acute stress response, the body's capacity to initiate inflammation and fever rapidly increases (e.g., through the cytokine *interleukin 6* [49]). Under normal circumstances, cortisol plays a role in suppressing this rapid inflammatory response, preventing it from overshooting [22]. This suppressive effect of cortisol is adaptive and fully reversible (*allostatic strain*). Yet, when cortisol levels are high for a longer period of time, the immune system becomes less sensitive to cortisol; that is, prolonged high cortisol causes *glucocorticoid receptor resistance* [50–52]. People who have glucocorticoid receptor resistance, in turn, are more vulnerable to *nonresolving inflammation*, a condition that may progress into many disease states, including arthritis, asthma, and cancer [53, 54].

Third, cortisol can cause disease through its actions on the metabolic system. Cortisol increases circulating glucose, which helps people to sustain their ongoing attempts to deal with stressors [22]. This process is adaptive and fully reversible (*allostatic strain*). However, when cortisol levels stay high, a cascade of physiological processes takes off, and these processes can together cause damage. Perhaps most notably, although the interaction between cortisol and insulin is complex, it is fair to say that cortisol has the potential to disrupt insulin functioning [55]. Under normal circumstances, insulin prompts cells to take up glucose, so that the glucose can be used for glycolysis, the process that energizes cells. Yet, when cortisol levels continue to be high, maintaining the high levels of circulating glucose, more and more insulin is required to achieve the same results as before; that is, high cortisol can cause *insulin resistance* [56]. Moreover, sustained high levels of cortisol, facilitate the accumulation of abdominal fat [57, 58]. These two processes–the development of insulin resistance and the accumulation of

abdominal fat (*allostatic load*)–can together progress into *diabetes type 2* [55]. This is a disease state characterized by symptoms such as thirst, hunger (also after eating), unexplained weight loss, fatigue, and headaches.

As we aimed to develop a parsimonious model, we attempted to capture only the essence of these cortisol-to-disease mechanisms in our model's formulas. In particular, we modeled the link between cortisol and disease as follows: (a) the cortisol response puts a burden on the body that is, in principle, fully reversible (*allostatic strain*); (b) however, when this burden is sustained, lasting tissue damage occurs (*allostatic load*); (c) such lasting tissue damage can lead to *disease* (see Fig 1).

To further illustrate how we modeled the cortisol-to-disease pathway, we note that our approach is similar to the so-called *rubber band analogy* (e.g., [59–61]). The rubber band analogy holds that the pathway from stress to disease is analogous to how rubber bands can get damaged when they are stretched. Also in this domain, three stages can be distinguished: (a) the rubber stretches, but it can regain its original shape (this is called *elastic deformation*); (b) the rubber stretches further, and it can no longer return to its original shape (this is called *plastic deformation*); (c) the rubber tears (this is called *failure*). These three stages are akin to *allostatic strain, allostatic load*, and *disease*, respectively. That is, in our model, the pathway from stress to disease is analogous to how a rubber band, when stretched, subsequently undergoes elastic deformation, plastic deformation, and failure.

We will evaluate our model by examining its ability to reproduce the relationship between work stressors and disease, which is well-supported by previous empirical work. Specifically, the highest level of evidence is available for the relationship between *job strain* (the combination of high job demand and low job control) and cardiovascular disease [62–65]. That is, a meta-analysis [62], which included over 600,000 participants from 27 cohort studies, revealed a positive relationship between work strain and cardiovascular disease with a relative risk of 1.33 (95% confidence interval: 1.19, 1.49). As *job strain* can be seen as a close proxy for the prevalence of *work stressors*, we will use this previously-observed, positive relationship as a benchmark for our model predictions.

## In a computational model

We refer to the current burden on physiological systems as *allostatic strain*, S. Like rubber bands, physiological systems usually return to their original state. In line with this principle, we describe change in *allostatic strain*, $\frac{dS}{dt}$, as follows:

$$\frac{dS}{dt} = -\rho_R S_t + C_t \tag{2}$$

In Eq 2, $S_t$ represents *allostatic strain* at a given time point; $\rho_R$ is the *recovery coefficient*, which determines the speed of recovery proportional to the current level of allostatic strain; and, $C_t$ is the current force exerted on the physiological system (represented by cortisol in our model).

When allostatic strain exceeds beyond a threshold, we assume that lasting damage occurs. We refer to lasting damage as *allostatic load*, L. The *allostatic threshold*, $\varepsilon_L$, is the point where allostatic load starts to form. We can write:

$$\frac{dL}{dt} = \begin{cases} 0, & S_t < \varepsilon_L \\ S_t - \varepsilon_L, & S_t \geq \varepsilon_L \end{cases} \tag{3}$$

Eq 3 states that allostatic load does not change if the current allostatic strain, $S_t$, is lower than $\varepsilon_L$. However, when the current allostatic strain exceeds the allostatic threshold, allostatic load increases by the excess, $S_t - \varepsilon_L$.

When physiological systems are repeatedly stretched beyond the allostatic threshold, this will cause allostatic load to accumulate. We assume that when allostatic load passes the *disease threshold*, $\varepsilon_D$, people become diseased. We refer to people's *disease state* as $D$. We can write:

$$D_t = \begin{cases} \textit{False}, & L_t < \varepsilon_D \\ \textit{True}, & L_t \geq \varepsilon_D \end{cases} \tag{4}$$

Eq 4 states that the *disease state*, $D_t$, is False (i.e., the person is not diseased) if allostatic load, $L_t$, is below the disease threshold $\varepsilon_D$. Conversely, $D_t$ is True (i.e., the person is diseased) if $L_t$ exceeds $\varepsilon_D$.

Together, Eqs 2–4 describe how cortisol can cause disease. Fig 8 illustrates what variation of allostatic strain, allostatic load and disease can be expected in our model simulations. The reader should note that because choices of $\rho_R$, $\varepsilon_L$ and $\varepsilon_D$ are arbitrary, the absolute time until disease in our model is meaningless. Rather, the relative time until disease between simulated individuals is what is of interest in our simulations.

## Simulations

We performed simulations based on our model, as defined in Eqs 1–4, using several values for the parameters $\rho_R$, $\varepsilon_L$ and $\varepsilon_D$ (code available in S3 Script). Specifically, we used three values for each parameter, yielding 27 combinations of parameter values. As the model is not scaled to resemble real-life observations (e.g., there is no clear-cut measure for *allostatic strain* and *allostatic load*), the selected parameter values were arbitrary. We selected these parameter values after exploration, such that, in each simulation, at least some people would become diseased and at least some people would not. Apart from this restriction, we selected parameter values over the full possible range.

As model input, cortisol time courses were created exactly as described in Study 1 (Table 1), apart from two minor variations. First, we aimed to increase the precision of detecting differences that exist in the simulated population, without needing to increase the number of simulated people. Thus, instead of simulating a population with a variation in work impulses that is described by a Gamma distribution, we simulated individual variation in work impulses from a uniform distribution over the same range of work impulses. In other words, instead of creating a population with a variation in work impulses that may resemble real-life populations, we now simulated a population with work impulses that are evenly distributed. Apart from increasing the precision of detecting effects, this variation does not otherwise influence the outcome of the simulations. Second, to save computational time, we simulated 5,000 instead of 10,000 people per parameter combination. Like before, for each parameter combination, we simulated both a scenario where night impulses and work impulses were uncorrelated (Scenario I), as well as a scenario where they were correlated (Scenario II).

## Simulation results

Fig 9 shows the results of the simulations. The figure shows the simulated relative risk of people becoming diseased, as a function of the number of work impulses. S1 Appendix includes a plot of accumulated allostatic load against the average amount of daily work stressors.

Results indicate that, in case of Scenario II, where work and night impulses are correlated, the model was robust to variation of the parameters $\rho_R$, $\varepsilon_L$ and $\varepsilon_D$. Irrespective of the variation

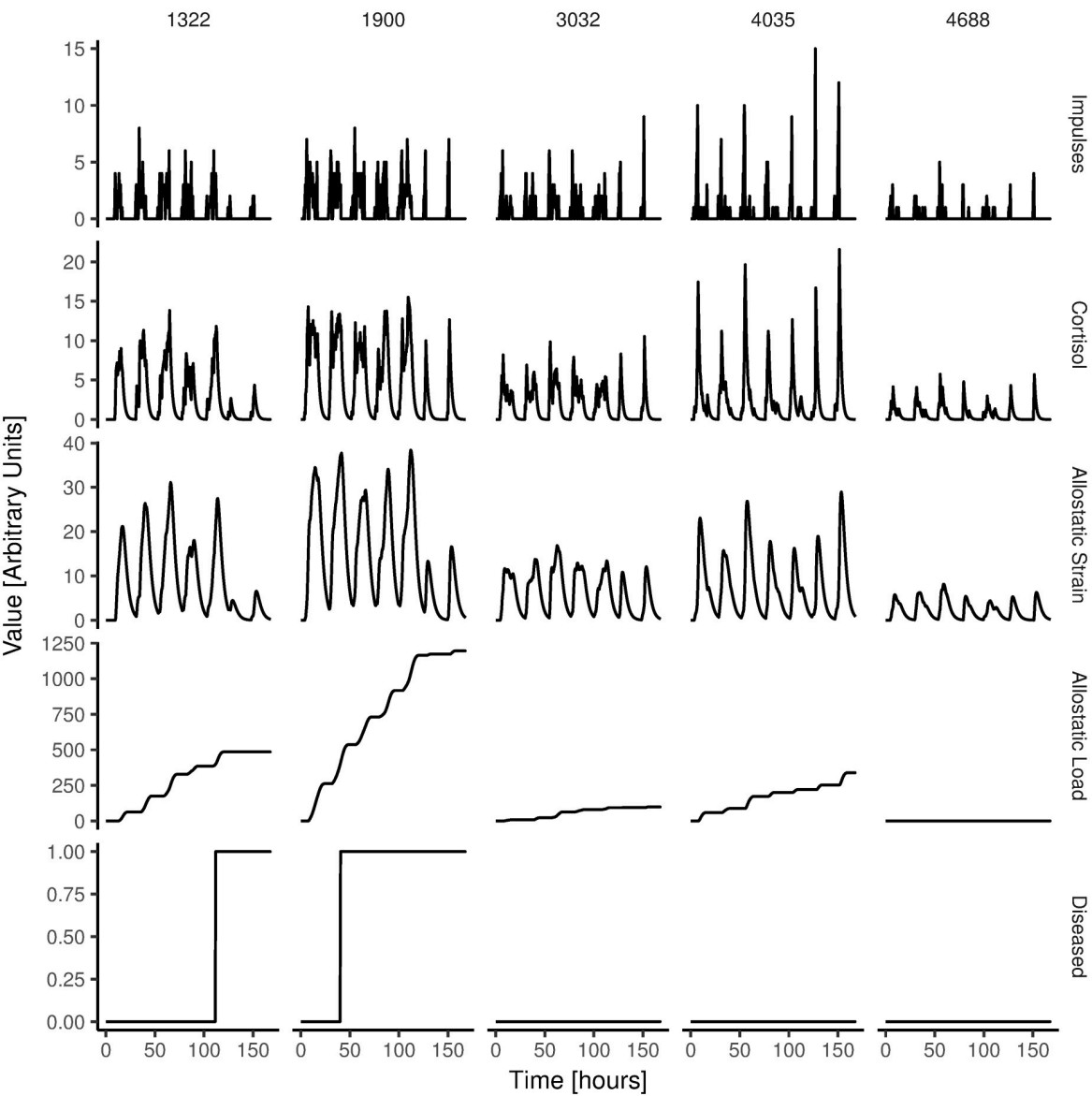

**Fig 8. Illustration of variation of model parameters.** Sample of seven days showing variation on all model variables from five random simulated people.

in the parameter values, we found a positive relationship between the frequency of work impulses and the simulated risk of becoming diseased (note that, because choices of $\rho_R$, $\varepsilon_L$ and $\varepsilon_D$ are arbitrary, the absolute effect sizes are meaningless). Also, we observed little variation in the predicted effect sizes, regardless of the specific parameter values (Fig 9b). These results are in line with the well-established empirical relationship between work stressors and disease [62], which we aimed to reproduce.

We did not observe this same robustness when we examined Scenario I. Although we did observe a positive relationship between work impulses and the simulated risk of disease for a number of parameter combinations, this relationship did not emerge in all cases (Fig 9a). Also, further exploratory analyses revealed that the cases where a positive relationship was observed

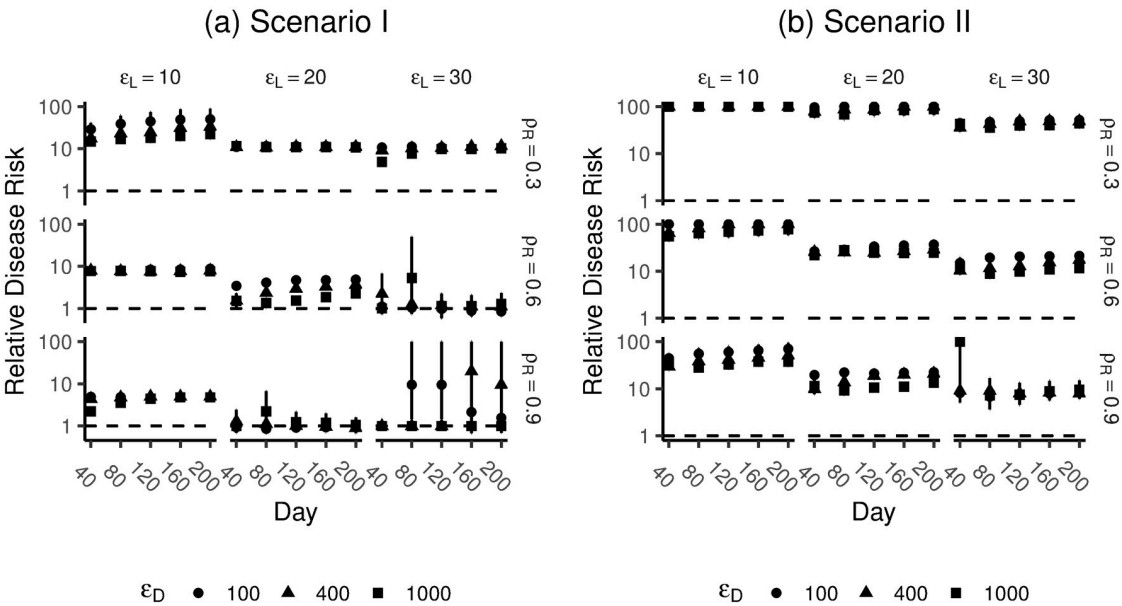

**Fig 9. Simulation results of relative risk of becoming diseased at various time points in the simulation.** Scenario I (a; no correlation between work and night impulses) Scenario II (b; correlation between work and night impulses). The odds ratios are calculated from the standardized averages of work impulses (i.e., the effect sizes represent one SD difference in the average number work impulses). Odd ratios greater than 100 are presented as 100.

for Scenario I, an unrealistically high proportion of simulated people had become diseased (see S1 Appendix).

## Discussion

At least when we assume that work and night impulses are correlated (Scenario II), which we would expect if people anticipate the events of the following workday [66, 67], the model successfully produced a positive correlation between work impulses and disease. This relationship, which is in line with the relationship between work stress and disease that is often observed in empirical literature [62], holds for a wide range of values of the parameters $\rho_R$, $\varepsilon_L$ and $\varepsilon_D$, supporting the robustness of the model. However, we did not find a robust relationship between work impulses and disease if work and night impulses were uncorrelated (Scenario I).

With caution, this set of findings raises the possibility that acute responses to work stressors may play only a limited role in the development of disease. Rather, it appears that, in comparison to the response to acute work stressors (the work impulses in our model), the contribution of the night impulses is so pronounced, that the cortisol spikes from the acute stressors during the working day contribute relatively little to the accumulation of allostatic load. In line with this observation, exploration of individual time courses of allostatic load show that the increase in allostatic load is mainly seen at the time point of the morning peak (for an illustration, see Fig 8).

### Study 3: New predictions for variations in workweek configurations

To recap, so far, we developed a computational model of work stress and disease. We compared the predictions from model simulations against empirical data. We found that, if we assume that night impulses (i.e., activity of the HPA axis during the night) and work impulses

work impulses (i.e., activity of the HPA axis due to work stressors) are correlated, our model can reproduce results from empirical research. Based on this finding, we conclude that our model is an appropriate model of the relationship between work stress and disease.

We will now proceed with generating new predictions, based on our model. In particular, we examine the effect of how working hours are distributed over the week. We do so by comparing the consequences of six workweek configurations on the development of disease (Table 2). We selected workweek configurations based on three observations. First, around the world, employees typically work 30–50 hours per week on average, usually on consecutive days (e.g., Monday-Friday [68]). Second, some organizations have experimented with *compressed working weeks*, i.e., working weeks in which working hours are concentrated in a limited number of days (e.g., 4), while the total number of working hours is unchanged (e.g., [69]). Third, somewhat anecdotally, some organizations have experimented with splitting up the working week in two parts, e.g., by introducing work-free Wednesdays [70]. Repeatedly, in some parts of Europe (e.g., France and The Netherlands), school children have Wednesday afternoons off, likely motivating parents that work part-time to negotiate to not work on Wednesdays.

## Simulations

In all simulations, we used a standard working week (Monday to Friday, 8 hours per day), like we used in all previous sections (Table 1), as a benchmark. Consistent with popular work-free days in parts of Europe, Wednesdays and Fridays have been selected as work-free days in some of the simulations. All configurations have additionally been simulated with ten hours longer and shorter, to investigate whether patterns that might be observed are consistent as weekly working hours increase or decrease. Table 2 presents all workweek configurations that we examined with separate simulations; Table 3 describes the parameter values that were identical across all simulations. Note that these values are identical to the values that we used previously, except that we used only one combination of the parameters $\rho_R$, $\varepsilon_L$ and $\varepsilon_D$ (simulations for other parameter combinations are reported in S1 Appendix). Also, based on our previous findings (see Study 2), in all simulations that follow, we assumed a correlation between night and work impulses.

## Simulation results and discussion

Fig 10 provides an overview of all simulation results, showing the predicted effects of all workweek configurations in Table 2 on the development of disease. The results show a clear pattern. Specifically, for 40h working weeks, we found an increase in predicted disease risk for configurations #3 and #4, in which working hours are concentrated in a limited number of days (a so-called *compressed working week* [69]). In contrast, the results show a decrease in disease risk in configurations #5 and #6, in which working hours are spread out over the week. We found no effect of configuration #2, in which work-free days are distributed throughout the week, rather than chunked together in a weekend. This pattern was persistent across the various numbers of working hours per week that we simulated. That is, regardless of we simulated 30h, 40h, or 50h workweeks, compared to configuration #1, (a) a compressed working week always increased the disease risk, (b) a spread-out working week reduced the disease risk, and (c) having work-free days on Wednesdays instead of Saturdays did not markedly impact disease risk. In sum, our model predicts that spreading out working hours over more days, rather than concentrating working hours in less days, helps prevent the development of disease.

We note another, related regularity from these simulation results (Fig 10). That is, independently of the number of working hours per week, there is a clear, positive relation between the

**Table 2. Workweek configurations that we explored with simulations.**

| Configuration | Working days | Days worked (per week) | Hours worked (per day) | | |
|---|---|---|---|---|---|
| | | | 30h/week | 40h/week | 50h/week |
| #1 | Mon-Fri | 5 | 6h | 8h[a] | 10h |
| #2 | Mon-Tue and Thu-Sat | 5 | 6h | 8h | 10h |
| #3 | Mon-Tue and Thu-Fri | 4 | 7h30m | 10h | 12h30m |
| #4 | Mon-Tue and Thu | 3 | 10h | 13h20m | 16h40m |
| #5 | Mon-Sat | 6 | 5h | 6h40m | 8h20m |
| #6 | Mon-Sun | 7 | 4h17m | 5h43m | 7h09m |

[a]Benchmark for all other configurations.

number of working hours per day and the predicted disease risk. In line with this regularity, our simulations suggest that 50-hour working weeks can be relatively healthy–i.e., healthier than the standard 40-hour/5-day working week–as long as these 50 hours are distributed over 7 short days (7.09 hours per day). Similarly, our simulations suggest that 30-hour working weeks can be relatively unhealthy, when these 30 hours are cramped into 3 long days (10 hours per day). In sum, although the total number of weekly working hours generally matters a lot (i.e., on average, disease risk is higher when people work more hours per week, Fig 10), our simulations suggests that the increased disease risk associated with many working hours per week can be mitigated, at least in part, by distributing these working hours over more days.

These simulation results speak to a recent set of trials conducted in Sweden, where employees were allowed to work 30-hour/5-day working weeks (instead of 40-hour/5-day working weeks; no pay cut applied). Although results of these trials were reported informally–to our knowledge, mainly in media publications (e.g. [71, 72])–our simulation results can be used to make new predictions about whether such trials are likely to support employees' long-term health. Specifically, we predict that the Swedish 30-hour configuration (i.e., configuration #1; 30-hour/5-day) is indeed healthier than its 40-hour counterpart (i.e., configuration #1;

**Table 3. Parameter settings that were held constant between the simulations of varying worktime configurations.**

| Parameter | Value |
|---|---|
| Sampling frequency | 2 |
| People simulated | 5,000 |
| Days simulated per person | 200 |
| Night impulses, quantity | $X \sim \Gamma(k = 3, \theta = 14)$ |
| Night impulses, moment | $X \sim T_{wake} - EXP(\lambda = 1)$ |
| Work impulses, quantity | $X \sim \mathcal{U}(a = 1, b = 50)$ |
| Work impulses, moment | $X \sim \mathcal{U}(a = T_{ws}, b = T_{we})$ |
| Cortisol decay constant ($\rho_C$) | $X \sim \mathcal{N}(\mu = .52, \sigma^2 = .05^2)$ |
| HPA axis scaling constant ($\kappa_{HPA}$) | 2.20 |
| Cortisol response delay, $\tau$ | 30 min |
| Wake time, $T_{wake}$ | 7:00AM |
| Work start, $T_{ws}$ | 8:30AM |
| Elasticity constant, $\rho_R$ | 0.6 |
| Allostatic threshold, $\varepsilon_L$ | 20 |
| Disease threshold, $\varepsilon_D$ | 400 |

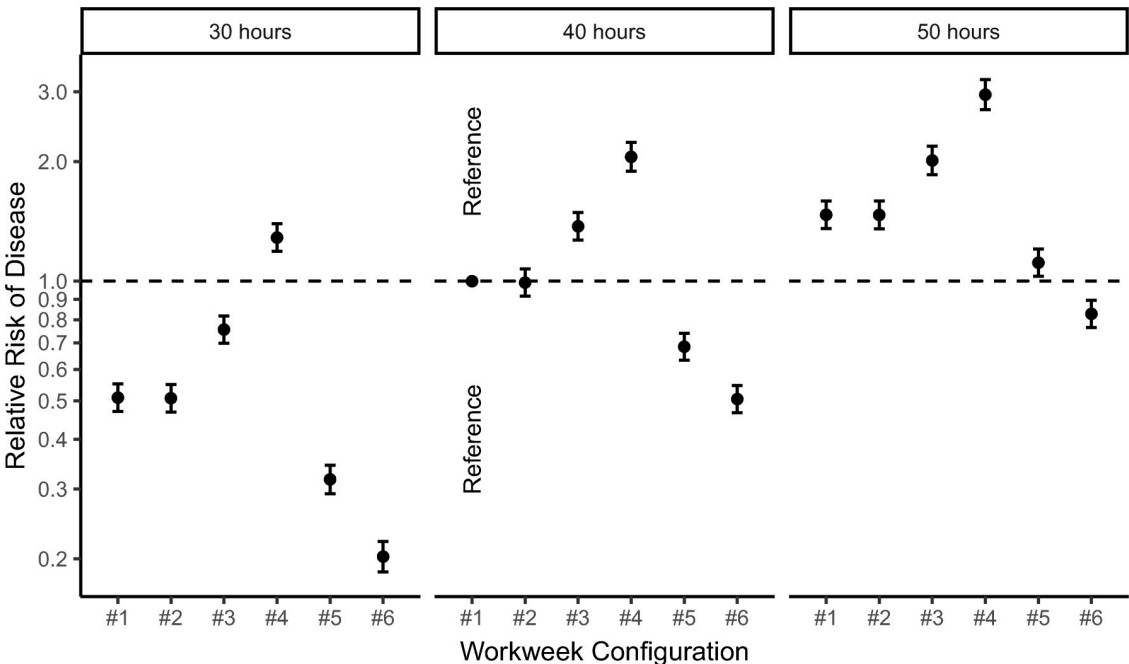

**Fig 10. New predictions based on our model.** The plot represents the relative risk of developing disease, as a function of different workweek configurations. All predictions are relative to a standard working week (i.e., configuration #1: 5 working days of 8 hours, Monday to Friday). See Table 2 for an explanation of all workweek configurations that we examined.

40-hour/5-day), but that there may be even healthier ways to distribute 30 hours over a working week (i.e., configuration #5, 30-hour/6-day; and configuration #6, 30-hour/7-day).

## General discussion

In this research we aimed to explain how work stress can cause disease, by developing a new computational model. In Study 1, we found that a model that defines cortisol dynamics in two linear relationships (i.e., release of cortisol is proportional to the number of stressors; decay of cortisol is proportional to the current cortisol level), can reproduce the characteristic shape of the cortisol day curve very well [28, 36, 37]. In Study 2, we found that we could reproduce the previously-reported relationship between work stress and disease [62], by formalizing the central predictions from *allostatic load theory*. In Study 3, based on our model, we formulated several novel predictions about the relationship between work stress and disease.

### Implications

Our research makes several contributions to the literature on the link between work stress and disease. First, we provide a first formalization of the core theoretical ideas in this domain, which previously existed only in the form of verbal theory. Thus, our computational model *contributes to the falsifiability* of these existing core ideas. Indeed, in our model, there are no implicit assumptions and there is no flexibility in interpretation of how parameters interact. As a result, going beyond previous theories, we can be explicit as to what observations would be needed to falsify our model: any pattern of data that is inconsistent with our predictions (Fig 10), would falsify our model. In that case, our model would need to be improved or replaced by an alternative model.

Second, by developing our model, we gained *new insights about the mechanisms* that together form the causal chain between work stress and disease. For example, we learned that the key logic of allostatic load theory holds when its predictions are scrutinized using a computational model, suggesting that allostatic load theory provides, in principle, a set of premises that is useful to understand how work stress can cause disease. Perhaps more importantly, we found that to show a robust link between work stress and disease, we need to assume that night impulses into the HPA axis are correlated with work impulses to the HPA axis. This finding is consistent with the idea that the morning peak, at least in part, stems from the *anticipation* of upcoming stressors [31, 67]. Our findings suggest that the acute cortisol response to work stressors by themselves, does not strongly contribute to the development of disease. Instead, work stressors may cause disease as their anticipation (e.g., during the night before) augments the morning peak. Interestingly, in line with the latter idea, several studies show that cardiovascular incidents are indeed most frequent in the first hours after awakening (e.g. [73, 74]).

Third, through simulations, we made several *new predictions* about the relationship between working hours and the development of disease. In essence, our model predicts that spreading out working hours over more days, rather than concentrating working hours in less days, prevents the development of disease. Related to this, our model predicts that the number of working hours per day (more so than the number of working hours per week) strongly predicts the development of disease. In the following paragraphs, we will discuss these predictions against the background of existing literature.

## The compressed working week

Our predictions are related to previous research on the concept of the compressed working week. The *compressed working week* is defined as a working week in which the weekly number of working hours is completed in fewer than five working days [69]. Generally, researchers hypothesized that the compressed working week should benefit employees' well-being via the restorative effects of having longer weekends. In particular, longer weekends should enable employees to pursue more leisure activities; to invest more time in social relationships [69, 75, 76]; and to detach and recover more strongly from work [77]. However, findings are mixed: some studies show that compressed working weeks increase well-being (or job satisfaction [69, 75]), but other studies show negative effects [76].

How to reconcile these mixed findings with our prediction that compressed working weeks can cause disease in the long run? We should point out that it is questionable whether these previous studies are relevant to our prediction at all. After all, they do not directly speak to employees' long-term physical health. Nevertheless, if we sidestep this important issue, these mixed findings are inconsistent with our predictions. This apparent inconsistency may be explained by one specific assumption in our model simulations, i.e., the assumption that the number of daily work stressors is linearly related to the number of working hours. In other words, we assumed that longer work days cause proportionally more impulses into the HPA axis. It is possible, though, that employees experience longer working days (in compressed working weeks) as an expression of *control* over their working hours, mitigating the negative effect from work stress. Indeed, the perception of control over working hours is a powerful mechanism to reduce the impact of stressors [78]. In future research, our model (and our simulation approach) can be used to further explore this issue.

Independent of this assumption regarding longer work days, we should note that our simulations still do not suggest a special health benefit from having longer weekends. In particular, in our simulations, increases in *allostatic load* happened on each individual workday, rather

than that increases in allostatic load gradually developed throughout the working week. So, our simulations suggest that it does not matter whether two (or more) work-free days are chunked together in a weekend. If research proves otherwise–i.e., if data would convincingly show that, all else being equal, longer weekends protect long-term health–this would falsify our model, and an improvement to our model would be necessary (e.g., we would need to add a parameter).

### Chronic stress and the cortisol morning peak

Our simulations suggest a relationship between the cortisol *morning peak* and the development of disease. A larger morning peak, so we predict, can cause disease in the long run. We assume that the morning peak stems from night impulses into the HPA axis, which are in part due to the anticipation of the upcoming day.

Interestingly, previous research suggests that the amplitude of the morning peak changes when people chronically experience stress [79]. In particular, some studies report that chronic stress leads to *reduced* responsiveness of the HPA axis, causing *hypocortisolism*, of which a lower morning peak is a key symptom [66]. Hypocortisolism has been reported in people who have previously experienced sustained or intense stress, such as people with post-traumatic stress disorder [35] and perhaps people with severe burnout [80]. Speculatively, stress-induced hypocortisolism may function to protect people from further physical damage [66]. After all, lower cortisol levels lead to less sustained burden on the cardiovascular, metabolic, and immune systems, potentially preventing disease.

In sum, (a) morning peaks may contribute to disease, but (b) the central nervous system may suppress morning peaks after exposure to sustained stress, and (c) such suppression, which happens in people with burnout, may protect people from further damage. Connecting these three ideas, it is possible to formulate a novel perspective on the nature of burnout. In particular, burnout is traditionally conceptualized as a highly aversive syndrome that emerges from known work-related conditions [81], that can be best treated in a way that is tailored to the client (e.g., [82]), making use of established treatments for depression [83]. We suggest that burnout can also be seen as an adaptive state that shields people from developing potentially lethal conditions, such as myocardial infarctions. This alternative conceptualization (burnout as a protective mode of functioning) is not necessarily better than the original (burnout as a work-related syndrome related to depression), but we suggest that it is potentially productive to consider both conceptualizations together in future research.

### Limitations and future directions

As we strived to develop a minimal model, with only few parameters, we have simplified reality in several ways. Each of these simplifications constitutes a potential limitation of our model, as they may cause our model to have one or more blind spots.

First, perhaps reflecting the most rigorous simplification of reality in our study (for a discussion, see [84]), we used cortisol as the only indicator of allostasis. However, allostasis is thought to involve a host of regulatory systems, including not just the HPA axis, but also the sympathetic branch of the autonomic nervous system [5, 67]. We chose to focus only on cortisol, because the role of cortisol in stress is very well-studied, which allowed us to calibrate our model to large samples [37]. Moreover, cortisol affects several regulatory systems, including the sympathetic nervous system. Still, our reductionistic approach constitutes a departure from previous, broader attempts to operationalize allostasis and allostatic load [85], and future research is needed to examine whether this simplification can be justified.

Second, our model only considers variation in the *frequency* of work stressors, not their *duration*. In fact, we assumed that all impulses into the HPA axis have the same duration. Importantly, despite this limitation, our model was well able to reproduce core characteristics of the cortisol day curve (Figs 5–7). So, on first sight, it seems that adding variation in the duration of stressors would not lead to a big improvement in our model. Nevertheless, real-life stressors vary in their duration, and stressors may differ in the duration in the physiological response that they trigger (e.g., stressors may have a sustained impact if they lead to perseverative cognition, [32]). More broadly, at least in its current form, our model cannot be used to make predictions about the impact of between-stressor differences (e.g., whether stressors stem from work vs. from other sources; whether stressors are short vs. long). We acknowledge that these differences exist, and that they may matter.

Third, for any simulated person, we assumed that stressors are equally likely on every working day (e.g., we assume stressors are equally likely on Mondays or Fridays). We should note that this assumption is a simplification of reality. For example, prior research suggests that the cortisol dynamics of a given day are influenced by stressors from the previous day [86]. Moreover, cortisol responses may become somewhat stronger with consecutive working days [87]. We acknowledge that such patterns may affect the predictions made by our model. Extensions of our model would be necessary to explore the importance of this issue.

Fourth, in our simulations, we assumed the same values for all simulated individuals for the parameters $\rho_R$ (recovery coefficient), $\varepsilon_L$ (allostatic load threshold) and $\varepsilon_D$ (disease threshold). As some of our parameters were already unscaled (i.e., they were not directly linked to quantities measured in real life, e.g., this was the case for $S$, allostatic strain, and $L$, allostatic load), we chose not to vary values for $\rho_R$, $\varepsilon_L$ and $\varepsilon_D$ in order to avoid adding additional arbitrary assumptions of inter-individual variability. However, it unlikely that this simplification is realistic; in fact, it is plausible that inter-individual variation in health and healthy lifestyle are represented in these parameters. Future research is needed to examine the impact of such variation.

Fifth, we should note that the fact that we included unscaled parameters prevented us from estimating precise effect sizes and time-to-disease values. However, we should note that our model does allow for making relative comparisons, to determine which scenario has a higher risk of leading to disease. If researchers succeed in determining meaningful values for some of the parameters (e.g., $S$ and $L$), it will be possible to estimate effect sizes of different workweek configurations and to estimate time-to-disease values, which we think would present a valuable and meaningful addition to the current model. Although this was not the aim of this study, we do feel this constitutes an interesting direction for future research. To this purpose, researchers may find work by Gruenewald et al. [88] valuable.

Sixth, mechanisms other than the ones we modeled, can potentially explain the relation between work stress and disease. Perhaps most importantly, our model does not consider the possibility that stress may lead to disease through poor health decisions (e.g., unhealthy eating [89], smoking [90], alcohol use [91]). Moreover, some research suggests that gradual changes in the circadian cortisol profile after a long period of chronic stress (specifically, a flattened profile with blunted morning cortisol peaks and elevated cortisol levels during the rest of the day) contributes to stress related disease (e.g., [92]). Our computational model models neither of these mechanisms. However, as we mentioned in the introduction, rather than providing a comprehensive model, we aimed to create and test a parsimonious and straightforward model of the stress-disease relationship. This being said, we do envision that modeling these alternative mechanisms will lead to interesting insights. In particular, a model that includes the health behavior route may lead to interesting, counter-intuitive predictions, as such a model is likely to involve complex dynamics.

Taken together, we see several avenues for future research. Most importantly, a next step could be to collect empirical data (or to re-use existing empirical data) to critically test our model's predictions. This would help examine whether our model's development is on the right track, or whether large changes are needed. However, even in its current form, our model can already be used to generate predictions about specific configurations of working weeks, beyond those covered in this paper. Finally, our model can be adapted or refined, e.g., by adding and/or removing parameters, to make new predictions about the role of job characteristics (e.g., worktime control, job control, social support, emotional and physical demands of work, task variety), and perhaps, to examine the long-term health implications of other aspects of working life (e.g., whether people can recover well at home; whether people encounter stressors for non-work-related reasons).

## Conclusion

The consequences of work stress are costly to individuals, organizations, and society at large. By formalizing the central assumptions of allostatic load theory, we were able to derive new hypotheses (about the effects of workweek configurations), which would have been difficult to derive through intuitive reasoning. At the same time, we contributed to the science of work stress by providing a model that is directly falsifiable. Reflecting on these results, we strongly feel that computational modeling is a fruitful approach not only for science, technology, and engineering, but also for occupational health. We hope this research may serve as a blueprint for making further progress in this important area.

## Supporting information

**S1 Appendix. Supplement figures and tables.**
(DOCX)

**S1 Fig. Comparison of simulations results and cohort data on AUC and wake-bed slope indices.**
(PDF)

**S1 Script. Cortisol simulation script.** Runnable R source code to simulate cortisol data.
(R)

**S2 Script. Cortisol visualization script.** Runnable R Markdown source code to create figures from the cortisol data from S1 Script.
(RMD)

**S3 Script. Full simulation script.** R source code for all simulations as discussed in the paper.
(R)

**S4 Script. Full visualization script.** R Markdown source code to create all figures from script S3.
(RMD)

**S5 Script. Figure script.** R source code to create all figures in the manuscript.
(R)

**S6 Script. AUC and slope script.** R source code to create S1 Fig.
(RMD)

**S1 Data. Figure data files.** Zip file with data used in S5 and S6 Scripts.
(ZIP)

## Acknowledgments

We thank two anonymous reviewers for their valuable input.

## Author Contributions

**Conceptualization:** Remco Benthem de Grave, Fred Hasselman, Erik Bijleveld.

**Data curation:** Remco Benthem de Grave.

**Formal analysis:** Remco Benthem de Grave.

**Methodology:** Remco Benthem de Grave, Fred Hasselman, Erik Bijleveld.

**Software:** Remco Benthem de Grave, Fred Hasselman.

**Supervision:** Fred Hasselman, Erik Bijleveld.

**Visualization:** Remco Benthem de Grave.

**Writing – original draft:** Remco Benthem de Grave, Erik Bijleveld.

**Writing – review & editing:** Remco Benthem de Grave, Fred Hasselman, Erik Bijleveld.

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
