## [Decision Letter · Decision Letter 0]

23 Oct 2021

PONE-D-21-10433

From work stress to disease: A computational model

PLOS ONE

Dear Dr. Remco Benthem de Grave,    

Thank you for submitting your manuscript to PLOS ONE.  As you will see from their comments below, the two expert reviewers were positive about the research. Nevertheless, the reviewers have recommendations for improving the submission. Thus, I invite you to resubmit your manuscript after addressing the comments below.

When revising your manuscript, please consider all issues mentioned in the reviewers' comments carefully: please outline every change made in response to their comments and provide suitable rebuttals for any comments not addressed. Please note that your revised submission will need to be re-reviewed.    

We look forward to receiving your revised manuscript.

Kind regards,

Brent Myers, Ph.D

Academic Editor

PLOS ONE

2. Please ensure that you refer to Figure 4 in your text as, if accepted, production will need this reference to link the reader to the figure.

**Comments to the Author**

1. Is the manuscript technically sound, and do the data support the conclusions?

Reviewer #1: Yes

Reviewer #2: Partly

2. Has the statistical analysis been performed appropriately and rigorously? 

Reviewer #1: Yes

Reviewer #2: Yes

3. Have the authors made all data underlying the findings in their manuscript fully available?

Reviewer #1: Yes

Reviewer #2: Yes

4. Is the manuscript presented in an intelligible fashion and written in standard English?

Reviewer #1: Yes

Reviewer #2: Yes

5. Review Comments to the Author

Reviewer #1: Review of PONE-D-21-10433

Though not criteria for PlosOne, I think this paper is quite important and would hopefully be impactful to the scientific community. I also thought the paper met the criteria for PlosOne. Here I provide comments that might improve on both types of criteria, essentially in order of appearance.

1. P. 2, line 10. I would replace “explain” with “represent”. This model is a representation of an existing explanation (as most are and there is nothing wrong with that).

2. P. 3, lines 42-43. The statement here is correct, but most models, including this one, are interpretations of theory. Thus, models do not strictly test theory either. Models test a rigorous representation of theory.

3. P. 3, lines 50-51. Researchers need to make interpretations regarding measurement and manipulations of constructs in the model. Modeling reduces but does not eliminate interpretation threats to valid conclusions. Note on lines 63-64 you refer to approximate empirical observations. That is what I am talking about.

4. P. 4, line 98. I think you mean “variable” where you say “sample”.

5. Figure 1. For the dynamics of cortisol level and allostatic strain you have a feedback loop with 1-rho(subscript C or R), respectively. Yet, Equations 1 and 2 do not have the constant, one, in them. Moreover, scaling would matter such that 1-rho in the figure implies that the dynamic process is a growth one, though at a rate inversely related to current level (i.e., if rho(C) times C < 1). To be consistent with equations and description, suggest “1-” be dropped from the triangles and “-“ should replace “+” in circles. You also do not make explicit the very reasonable assumption represented in the equations that decay rate is a function of level. Finally, not sure what loop for allostatic load is about.

6. Figure 3. X-axis is not clear. I guess we are looking at two days (48 hours).

7. P. 7, line 234. Seems referencing second panel in Figure 4, not 5a.

8. P. 8, line 238. Neither “only” is needed; certainly not both.

9. P. 8, line 240. Begin list without explaining list coming.

10. P. 11, line 358. What is the effect size metric? Odds ratio, relative risk?

11. P. 16, line 562. “on” should be “in”; line 568, remove second “to”; line 573, don’t you mean decrease, not increase?

12. P. 18, line 656. Not a sentence. Please rewrite. Line 665, change “to provide” to “providing”. Line 671, replace “the first next step should” with “a next step could”

13. P. 19, lines 689-690. Suggest replacing “not just for science, technology, and engineering – but that this approach can also help to make progress in the domain of occupational health” with “not only for science, technology, and engineering, but also for occupational health.”

In sum, I thought this was an excellent presentation of a computational model. Not too simple or too complex given the field. Great job!

Reviewer #2: I enjoyed reading your work on developing a computational model for work stress and disease. I had some concerns about the conclusions of the study, which I outline below, as well as potential ways to resolve these concerns:

1) In Study 1, you validated the computational model by comparing the cortisol pattern to those in previously published studies. I think this is a good approach; however, you don’t provide any specific statistical evidence to support this besides a visual examination of the figures. I’m wondering if you could compute the area under the curve, or the diurnal cortisol slope, for people in the published study and those in your simulation based on the computational model, and provide statistical evidence that these are similar.

2) Relatedly, in Study 3 you suggested that a more “spread out” work week has a greater impact on people’s disease risk than the number of hours they work. Can you provide evidence of this? You could simply regress the disease risk from your model on the work hours and work day configurations to show which is more important. Right now, I find the argument unconvincing and the figure looks (to me, at least) to show that work hours are just as if not more important than the timing. A statistical test of this would clear things up.

3) One thing I noticed from Figure 7 is that your model shows significantly less between-person variability in cortisol patterns than the empirical data you showed. In other words, the individuals in your model don’t seem to deviate too much from the mean, while in “real life” there is considerably more variation across people in their cortisol slopes over time. I guess I don’t know if this is a problem or not – on the one hand, the purpose of your model isn’t necessarily to provide individual-level (but rather population-level) descriptive information, so maybe it doesn’t matter. On the other, this points at the inherent “randomness” of physiological functioning, and your model doesn’t seem to be incorporating that well. Perhaps, to make it more realistic, there is a need to add some “noise” (i.e., random error) that might help better mimic real-life patterns? I’m interested to hear your thoughts on this.

4) I know the scaling of your computational model is arbitrary, but does it have to be? That is, could you introduce a scaling term that puts the simulated cortisol values on a serum cortisol scale? You could use published norms of “normal” cortisol ranges to develop this scaling parameter. I think this would facilitate the comparisons I am recommending in my first point, and provide more concrete practical implications to follow from your conclusions. For example, studies on allostatic load consider cortisol levels during the day above 21 (ug/g creatine) to put people at “high risk” for disease, so if your model can predict the conditions under which people will be in this high risk area it would be significantly more useful (Gruenewald et al., 2012). Of course, I could be missing the reason why you can’t scale the model to “actual” cortisol levels, and if that is the case feel free to ignore this suggestion.

5) A minor point, but can you provide a sentence or two justifying the specific workday combinations you tested in Study 3? The M-F is obvious (since I imagine most people work in this kind of schedule), but the others seem randomly selected. I don’t have any particular problem with any of the combinations, I just think some justification for the configurations you chose might be helpful.

6) Another minor point, and maybe I missed it, but does your model treat weekdays as interchangeable? In other words, is a Monday equivalent to a Wednesday in terms of work stress? I am not sure if this is a realistic assumption – for example, maybe people are less stressed on Mondays (because they’ve had the weekend to recover) than on Wednesdays (when stressors from the previous two days have been “building up”). I’m interested in hearing your thoughts on this, and can appreciate if introducing this type of thing in the model might make things overly complicated.

I hope you find my comments useful, and wish you the best of luck in revising your work.

Reference

Gruenewald, T. L., Karlamangla, A. S., Hu, P., Stein-Merkin, S., Crandall, C., Koretz, B., & Seeman, T. E. (2012). History of socioeconomic disadvantage and allostatic load in later life. Social science & medicine, 74(1), 75-83.

6. PLOS authors have the option to publish the peer review history of their article (what does this mean?). If published, this will include your full peer review and any attached files.

---

## [Author Response · Author response to Decision Letter 0]

17 Dec 2021

Dear Dr. Myers,

Thank you for providing us the opportunity to resubmit our manuscript with changes. 

Please find below our responses on both your and the reviewers' comments.

Looking forward to hearing your response.

Yours sincerely,

Remco Benthem de Grave

Response 1: Thank you. We have adjusted file naming in line with the PLOS ONE style requirements, changed the manuscript to double spacing, adjusted spacing on the title page, and added the initials of the corresponding author. 

2. Please ensure that you refer to Figure 4 in your text as, if accepted, production will need this reference to link the reader to the figure.

Response 2: Thank you for raising this point. We now added a reference in line 235. 

Reviewer #1: 

Though not criteria for PlosOne, I think this paper is quite important and would hopefully be impactful to the scientific community. I also thought the paper met the criteria for PlosOne. Here I provide comments that might improve on both types of criteria, essentially in order of appearance.

Response: Before going into details, we thank Reviewer #1 for their helpful comments and suggestions. 

1. P. 2, line 10. I would replace “explain” with “represent”. This model is a representation of an existing explanation (as most are and there is nothing wrong with that).

Response 1: We thank Reviewer #1 for this suggestion. We agree that “represent” works better here. We have adjusted the text as suggested.

2. P. 3, lines 42-43. The statement here is correct, but most models, including this one, are interpretations of theory. Thus, models do not strictly test theory either. Models test a rigorous representation of theory.

3. P. 3, lines 50-51. Researchers need to make interpretations regarding measurement and manipulations of constructs in the model. Modeling reduces but does not eliminate interpretation threats to valid conclusions. Note on lines 63-64 you refer to approximate empirical observations. That is what I am talking about.

Response 2 and 3: We thank Reviewer #2 for pointing this out. We agree that our explanation in the previous version of our manuscript was not fully accurate. In the new version, we stay away from the suggestion that computational models are totally free from interpretations by the researcher. Instead, we now merely suggest that verbal theory needs more interpretations than computational theory. Specifically, we now write:

“In order to perform any quantitative test of a verbal theory, e.g., using statistical analyses, researchers always need to make interpretations of the theory first, leaving much room for flexibility. It is thus difficult, and perhaps impossible, to rigorously test verbal theory [11].”

and 

“There is transparency in the dynamics that describe the relationship between parameters, limiting the assumptions that researchers have to make. Thus, computational models can typically be tested with greater rigor.”

4. P. 4, line 98. I think you mean “variable” where you say “sample”.

Response 4: We agree that “variable” is clearer here. We changed this as suggested. 

5. Figure 1. For the dynamics of cortisol level and allostatic strain you have a feedback loop with 1-rho(subscript C or R), respectively. Yet, Equations 1 and 2 do not have the constant, one, in them. Moreover, scaling would matter such that 1-rho in the figure implies that the dynamic process is a growth one, though at a rate inversely related to current level (i.e., if rho(C) times C < 1). To be consistent with equations and description, suggest “1-” be dropped from the triangles and “-“ should replace “+” in circles. You also do not make explicit the very reasonable assumption represented in the equations that decay rate is a function of level. Finally, not sure what loop for allostatic load is about.

Response 5: 

We made the following changes:

A) Reviewer #1 is absolutely right that the “1 – “ did not make sense in Figure 1. Many thanks for pointing out this mistake. We changed Figure 1 as suggested, so that it better reflects Equations 1 and 2. 

B) We now explicitly state that decay rate is proportional to the current level, both for cortisol and allostatic strain:

• We updated line 191 to: In this equation, a rise in the cortisol concentration that is proportional, by KHPA, to the neural impulses, It, delayed by tau. The decay of cortisol is proportional to the current cortisol concentration, Ct, by a decay constant, rhoC. 

• We updated lines 388-390 to: In Equation 2, St represents allostatic strain at a given time point; pR is the recovery coefficient, which determines the speed of recovery proportional to the current level of allostatic strain; and, Ct is the current force exerted on the physiological system (represented by cortisol in our model).

C) The loop referred to by Reviewer #1 reflected the assumption that allostatic load is cumulative. To make this cumulative aspect of our model clearer, we now replaced the loop by an integration symbol. We hope this is more intuitive. To avoid ambiguity, we also rewrote Figure 1’s caption, which now includes more explanation:

“Decay of cortisol and allostatic strain is assumed to be proportional to their respective current levels, as indicated by the feedback loops. When allostatic strain exceeds a threshold (εL), it causes permanent damage (allostatic load). Allostatic load is cumulative and non-reversible (hence the integration symbol). When allostatic load exceeds a threshold (εD), it ultimately causes disease. This chain of events is described in detail in the main text. HPA axis = Hypothalamus Pituitary Adrenal-axis.”

6. Figure 3. X-axis is not clear. I guess we are looking at two days (48 hours).

Response 6: We agree that the layout of the X-axis was not yet optimal in the previous version of our manuscript. We now changed the axis labels to represent two 24h cycles, to make clear that we are looking at two days of simulation, without having to look at the figure caption. The figure caption contains a more detailed explanation. 

7. P. 7, line 234. Seems referencing second panel in Figure 4, not 5a.

Response 7: Indeed, thanks. We have adjusted this as suggested.

8. P. 8, line 238. Neither “only” is needed; certainly not both.

Response 8: Yes, thanks. We have adjusted this too.

9. P. 8, line 240. Begin list without explaining list coming.

Response 9: We’ve added the sentence: “We compared the results from our simulations to results from previous empirical studies in three ways.” to introduce the list that follows.

10. P. 11, line 358. What is the effect size metric? Odds ratio, relative risk?

Response 10: We agree that this should be explicit. We now specify that we’re talking about relative risk on line 380.

11. P. 16, line 562. “on” should be “in”; line 568, remove second “to”; line 573, don’t you mean decrease, not increase?

12. P. 18, line 656. Not a sentence. Please rewrite. Line 665, change “to provide” to “providing”. Line 671, replace “the first next step should” with “a next step could”

13. P. 19, lines 689-690. Suggest replacing “not just for science, technology, and engineering – but that this approach can also help to make progress in the domain of occupational health” with “not only for science, technology, and engineering, but also for occupational health.”

Response 11, 12, and 13: We thank Reviewer #1 for these suggestions. We made changes in response to all of them. The sentence of line 705 now reads “Sixth, mechanisms other than the ones we modelled, can potentially explain the relation between work stress and disease.”

In sum, I thought this was an excellent presentation of a computational model. Not too simple or too complex given the field. Great job!

Response: We thank Reviewer #1 for this nice comment.

Reviewer #2

I enjoyed reading your work on developing a computational model for work stress and disease. I had some concerns about the conclusions of the study, which I outline below, as well as potential ways to resolve these concerns:

Response: We thank Reviewer #2 for their enthusiasm and their constructive review. 

1) In Study 1, you validated the computational model by comparing the cortisol pattern to those in previously published studies. I think this is a good approach; however, you don’t provide any specific statistical evidence to support this besides a visual examination of the figures. I’m wondering if you could compute the area under the curve, or the diurnal cortisol slope, for people in the published study and those in your simulation based on the computational model, and provide statistical evidence that these are similar.

Response 1: We agree that it would be better to add more substance to our comparison. As Reviewer #2 suggested, we have now performed calculations of AUC and the wake-to-bed slope. As we have access only to the aggregate data from the CIRCORT study by Miller et al. 2016 (i.e., we do not have data from individual subjects), we calculated the indices based on the different percentiles that Miller et al. present in their paper (i.e., 10th, 25th, 50th, 75th, 90th), and we then did the same for our simulations. We present results from this analysis in Figure S1, which shows further evidence that our simulations mimic the data very closely (i.e., for all indices, the points are on a very straight line). In the main text we have added the following: 

“Moreover, we present more detailed comparisons between this previous meta-dataset and our simulations in Fig S1. Specifically, for both the empirical data and our simulations, we quantified daily cortisol patterns as is often done in the literature (i.e., area under the curve with respect to the ground [40], AUCG; area under the curve with respect to the increase, AUCI [40]; and the wake-to-bed slope [41]). We then plotted the indices based on the empirical data against the indices based on the simulations (Fig S1), enabling us to further assess how well the simulations match the data, when considering well-established indices of diurnal cortisol dynamics.”

2) Relatedly, in Study 3 you suggested that a more “spread out” work week has a greater impact on people’s disease risk than the number of hours they work. Can you provide evidence of this? You could simply regress the disease risk from your model on the work hours and work day configurations to show which is more important. Right now, I find the argument unconvincing and the figure looks (to me, at least) to show that work hours are just as if not more important than the timing. A statistical test of this would clear things up.

Response 2: We absolutely understand Reviewer #2’s reservations. To clarify: It was never our intention to claim that the total number of hours in the working week was more important than how these hours are distributed over the week. Rather, we intended to claim that the way the working hours are distributed over the week matters, regardless of how many hours people work (within a normal range, e.g., 30, 40, or 50 h/week). We understand, however, that this was unclear in the previous version of our manuscript. So, we made several changes: 

A) We deleted a sentence where we accidentally made a misleading claim. We previously wrote: “In sum, although the total number of weekly working hours does matter (on average, disease risk is higher when people work more hours per week), our predictions suggests that the total number of daily working hours plays a more dominant role.” This sentence is gone now.

B) We rewrote lines 531 – 535 to describe our findings more precisely. For example, we now write “In sum, although the total number of weekly working hours generally matters a lot (i.e., on average, disease risk is higher when people work more hours per week, Fig 10), our simulations suggests that the increased disease risk associated with many working hours per week can be mitigated—at least to some extent—by distributing these working hours over more days.”

3) One thing I noticed from Figure 7 is that your model shows significantly less between-person variability in cortisol patterns than the empirical data you showed. In other words, the individuals in your model don’t seem to deviate too much from the mean, while in “real life” there is considerably more variation across people in their cortisol slopes over time. I guess I don’t know if this is a problem or not – on the one hand, the purpose of your model isn’t necessarily to provide individual-level (but rather population-level) descriptive information, so maybe it doesn’t matter. On the other, this points at the inherent “randomness” of physiological functioning, and your model doesn’t seem to be incorporating that well. Perhaps, to make it more realistic, there is a need to add some “noise” (i.e., random error) that might help better mimic real-life patterns? I’m interested to hear your thoughts on this.

Response 3: We thank Reviewer #2 for making this observation. We assume Reviewer #2 is referring to the lines in Fig 7a looking more jagged (or noisier) than the ones in Fig 7b. To us, there does not seem to me a lot more between-person variation in the real data (note that the individual Y-axes are scaled to the data). We hope we understood this comment correctly. In any case, we do have some thoughts about why Figure 7a vs. 7b look different, which we now describe in the paper:

“Third, when comparing individual-level cortisol time courses from empirical data and our simulations (Fig 7a vs. 7b), we note that there is more high-frequency variation in cortisol levels in real empirical data (i.e., lines in Fig 7a look more jagged). This difference can be explained from the fact that we used a relatively low sampling rate in our simulations. Specifically, in our simulations, stressors always had a duration of 30 minutes, whereas the duration of real-life stressors may vary considerably; they can be shorter than 30 minutes. Moreover, the measurement process that yielded the real-life data (i.e., participants had a catheter in their forearm; blood samples were taken every 10 minutes [28]) may have added some noise by itself. Although this difference between the real data vs. the simulated data may slightly distort variance around the effect sizes reported in later sections of this paper, we expect that the estimates themselves should not be biased by our simulations having less high-frequency noise. “

4) I know the scaling of your computational model is arbitrary, but does it have to be? That is, could you introduce a scaling term that puts the simulated cortisol values on a serum cortisol scale? You could use published norms of “normal” cortisol ranges to develop this scaling parameter. I think this would facilitate the comparisons I am recommending in my first point, and provide more concrete practical implications to follow from your conclusions. For example, studies on allostatic load consider cortisol levels during the day above 21 (ug/g creatine) to put people at “high risk” for disease, so if your model can predict the conditions under which people will be in this high risk area it would be significantly more useful (Gruenewald et al., 2012). Of course, I could be missing the reason why you can’t scale the model to “actual” cortisol levels, and if that is the case feel free to ignore this suggestion.

Response 4: We thank Reviewer #2 for thinking along with us, and for making suggestions to create the potential impact of our paper. 

A) To respond to the first point, we agree that the arbitrariness of the scale of the simulated cortisol levels does not necessarily mean that we cannot scale the values to real-life observations. Indeed, we have scaled our units to make simulated values similar to the values that were observed by Miller et al., ref 36; this becomes visible when comparing Fig 5a and 5b. Rather, the arbitrariness of our units stems from the fact that cortisol values were obtained independently from a specific measurement process. So, we could potentially scale our values to either blood, urinary or salivary cortisol—each of which reflects a different cascade of physiological processes (e.g., the cortisol peak in saliva after a stressor comes later than the peak in blood). Reporting a specific scale could suggest to readers that we calibrated our model to one of these modalities (e.g., by feeding our model with real stressor data), which we did not do. So, to stay away from making misleading claims, we call our scale ‘arbitrary’. 

B) With regard to Reviewer #2’s suggestion to define thresholds for people who are at risk (e.g., like Gruenewald et al., 2021, did), we fully agree that this would be very useful for practice. Unfortunately, beyond what we described at (A) above, there are several potential issues that would make the interpretation of such norms difficult. First, our model suggests that it is not peak cortisol levels that predict disease, but rather, the sustained elevated levels of cortisol, which cause allostatic strain to rise, which results in allostatic load and disease. Indeed, some previous work (e.g. Kumari et al., 2011, J. Clin. Endocrinol. Metab.) suggests it is not individual cortisol measurements, but larger trends (e.g., the diurnal slope) that predict health problems. So, in our model, it would not be so clear how a specific threshold (akin to 21 ug/g creatine) should be interpreted. We are not saying that the ambition to define threshold values is unrealistic, but we do think this would warrant a dedicated research project. Therefore, we see no adequate way to incorporate cut-offs we can trust for this publication. We see the relevance of the reference to Gruenewald et al. for this line of thinking, and we now refer to this work on line 703-704. 

5) A minor point, but can you provide a sentence or two justifying the specific workday combinations you tested in Study 3? The M-F is obvious (since I imagine most people work in this kind of schedule), but the others seem randomly selected. I don’t have any particular problem with any of the combinations, I just think some justification for the configurations you chose might be helpful.

Response 5: We thank Reviewer #2 for this suggestion, which makes sense to us. We have adjusted the text as follows: 

“We will now proceed with generating new predictions, based on our model. In particular, we examine the effect of how working hours are distributed over the week. We do so by comparing the consequences of six workweek configurations on the development of disease (Table 2). We selected workweek configurations based on three observations. First, around the world, employees typically work 30–50 hours per week on average, usually on consecutive days (e.g., Monday–Friday [68]). Second, some organizations have experimented with compressed working weeks, i.e., working weeks in which working hours are concentrated in a limited number of days (e.g., 4), while the total number of working hours is unchanged (e.g., [69]. Third, somewhat anecdotally, some organizations have experimented with splitting up the working week in two parts, e.g., by introducing work-free Wednesdays [70]. Relatedly, in some parts of Europe, school children have Wednesday afternoons off, likely motivating parents that work part-time to negotiate to not work on Wednesdays.”

 6) Another minor point, and maybe I missed it, but does your model treat weekdays as interchangeable? In other words, is a Monday equivalent to a Wednesday in terms of work stress? I am not sure if this is a realistic assumption – for example, maybe people are less stressed on Mondays (because they’ve had the weekend to recover) than on Wednesdays (when stressors from the previous two days have been “building up”). I’m interested in hearing your thoughts on this, and can appreciate if introducing this type of thing in the model might make things overly complicated.

Response 6: Indeed, the choice to assume the same probability of stressors on any specific working day is a simplification, which limits the accuracy of our predictions. It was our goal to develop a model that was able to replicate observations from large studies; it turned out that it was not necessary to include weekly trends in stress patterns in our model. We do feel that it would add to the paper to highlight this simplification. We therefore included it as a limitation:

“Third, for any simulated person, we assumed that stressors are equally likely on every working day (e.g., we assume stressors are equally likely on Mondays or Fridays). We should note that this assumption is a simplification of reality. For example, prior research suggests that the cortisol dynamics of a given day are influenced by stressors from the previous day [86]. Moreover, cortisol responses may become somewhat stronger with consecutive working days. We acknowledge that such patterns may affect the predictions made by our model. Extensions of our model would be necessary to explore the importance of this issue.” (Lines 678-685).

I hope you find my comments useful, and wish you the best of luck in revising your work.

Response: Absolutely. Thank you for your constructive input.

---

## [Decision Letter · Decision Letter 1]

2 Feb 2022

From work stress to disease: A computational model

PONE-D-21-10433R1

Dear Dr. Remco Benthem de Grave,

We’re pleased to inform you that your manuscript has been judged scientifically suitable for publication and will be formally accepted for publication once it meets all outstanding technical requirements.

Kind regards,

Brent Myers, Ph.D

Academic Editor

PLOS ONE

Additional Editor Comments (optional):

Reviewers' comments:

Reviewer's Responses to Questions

**Comments to the Author**

1. If the authors have adequately addressed your comments raised in a previous round of review and you feel that this manuscript is now acceptable for publication, you may indicate that here to bypass the “Comments to the Author” section, enter your conflict of interest statement in the “Confidential to Editor” section, and submit your "Accept" recommendation.

Reviewer #1: All comments have been addressed

Reviewer #2: (No Response)

2. Is the manuscript technically sound, and do the data support the conclusions?

Reviewer #1: Yes

Reviewer #2: Yes

3. Has the statistical analysis been performed appropriately and rigorously? 

Reviewer #1: Yes

Reviewer #2: Yes

4. Have the authors made all data underlying the findings in their manuscript fully available?

Reviewer #1: Yes

Reviewer #2: Yes

5. Is the manuscript presented in an intelligible fashion and written in standard English?

Reviewer #1: Yes

Reviewer #2: Yes

6. Review Comments to the Author

Reviewer #1: (No Response)

Reviewer #2: As before, I enjoyed reading your work describing a computational model of work stress. I think this is an interesting paper that has the potential to make an important contribution to the literature. I also applaud your responses to my inquiries, and think you have addressed them all well. I have one remaining question in line with a suggestion I made in the prior round; namely, could you regress the disease risk from your model on the work hours and work day configurations to show which of the two is more important in driving disease risk? I think this might provide important guidance to policymakers and employers, even if there might be some statistical issues with doing this (e.g., specific assumptions underlying the disease risk model). If you or the Editor think this isn't a good idea, though, I defer to your judgment. Great work!

7. PLOS authors have the option to publish the peer review history of their article (what does this mean?). If published, this will include your full peer review and any attached files.

Reviewer #1: No

Reviewer #2: No

---

## [Editor Report · Acceptance letter]

7 Feb 2022

PONE-D-21-10433R1 

From work stress to disease: A computational model 

Dear Dr. Benthem de Grave:

I'm pleased to inform you that your manuscript has been deemed suitable for publication in PLOS ONE. Congratulations! Your manuscript is now with our production department. 

Kind regards, 

on behalf of

Dr. Brent Myers 

Academic Editor

PLOS ONE